# Leucine rich amelogenin peptide prevents ovariectomy-induced bone loss in mice

**Naoto Haruyama**[1,2¤]*, **Takayoshi Yamaza**[3], **Shigeki Suzuki**[1,4], **Bradford Hall**[1], **Andrew Cho**[1], **Carolyn W. Gibson**[5], **Ashok B. Kulkarni**[1]*

**1** National Institute of Dental and Craniofacial Research, National Institutes of Health, Bethesda, MD, United States of America, **2** Section of Orthodontics and Dentofacial Orthopedics, Faculty of Dental Science, Kyushu University, Fukuoka-shi, Fukuoka, Japan, **3** Section of Molecular Cell Biology and Oral Anatomy, Faculty of Dental Science, Kyushu University, Fukuoka-shi, Fukuoka, Japan, **4** Department of Periodontology and Endodontology, Tohoku University Graduate School of Dentistry, Sendai, Miyagi, Japan, **5** Department of Basic and Translational Sciences, University of Pennsylvania School of Dental Medicine, Philadelphia, PA, United States of America

¤ Current address: Graduate School of Dentistry, Kyushu University, Fukuoka, Japan
* haruyama@dent.kyushu-u.ac.jp (NH); Ashok.Kulkarni@nih.gov (ABK)

**Data Availability Statement:** All relevant data are within the paper and its Supporting Information files.

**Funding:** This research was supported by the Divison of Intramural Research of the NIDCR, NIH.

## Abstract

Amelogenins, major extra cellular matrix proteins of developing tooth enamel, are predominantly expressed by ameloblasts and play significant roles in the formation of enamel. Recently, amelogenin has been detected in various epithelial and mesenchymal tissues, implicating that it might have distinct functions in various tissues. We have previously reported that leucine rich amelogenin peptide (LRAP), one of the alternate splice forms of amelogenin, regulates receptor activator of NF-kappa B ligand (RANKL) expression in cementoblast/periodontal ligament cells, suggesting that the amelogenins, especially LRAP, might function as a signaling molecule in bone metabolism. The objective of this study was to identify and define LRAP functions in bone turnover. We engineered transgenic (TgLRAP) mice using a murine 2.3kb α1(I)-collagen promoter to drive expression of a transgene consisting of LRAP, an internal ribosome entry site (IRES) and enhanced green fluorescent protein (EGFP) to study functions of LRAP in bone formation and resorption. Calvarial cell cultures from the TgLRAP mice showed increased alkaline phosphatase (ALP) activity and increased formation of mineralized nodules compared to the cells derived from wild-type (WT) mice. The TgLRAP calvarial cells also showed an inhibitory effect on osteoclastogenesis *in vitro*. Gene expression comparison by quantitative polymerase chain reaction (Q-PCR) in calvarial cells indicated that bone formation makers such as *Runx2*, *Alp*, and *osteocalcin* were increased in TgLRAP compared to the WT cells. Meanwhile, *Rankl* expression was decreased in the TgLRAP cells *in vitro*. The ovariectomized (OVX) TgLRAP mice resisted bone loss induced by ovariectomy resulting in higher bone mineral density in comparison to OVX WT mice. The quantitative analysis of calcein intakes indicated that the ovariectomy resulted in increased bone formation in both WT and TgLRAP mice; OVX TgLRAP appeared to show the most remarkably increased bone formation. The parameters for bone resorption in tissue sections showed increased number of osteoclasts in OVX WT, but not in OVX TgLRAP over that of sham operated WT or TgLRAP mice,

The funders had no role in study design, data collection and analysis, decision to publish, or preparation of the manuscript.

**Competing interests:** The authors have declared that no competing interests exist.

supporting the observed bone phenotypes in OVX mice. This is the first report identifying that LRAP, one of the amelogenin splice variants, affects bone turnover *in vivo*.

## Introduction

Amelogenin is the most abundant extra cellular matrix (ECM) protein secreted by ameloblasts during tooth development [1]. Amelogenin is secreted into the dental enamel matrix to form tooth enamel, a highly mineralized and the hardest tissue in the body. Amelogenin contributes more than 90% of the bulk of the organic matrix, which undergoes systematic proteolysis by specific proteases, matrix metalloprotease 20 (MMP20) and kallikrein (KLK4), during the tooth enamel mineralization process [2, 3]. Eventually, amelogenin is almost completely removed during the enamel maturation process [4].

Numerous mutations in the amelogenin gene have been identified in patients with the most common genetic disorder affecting enamel, amelogenesis imperfecta (AI) [5–8]. The targeted disruption of the amelogenin gene locus in mice also showed a hypoplastic enamel phenotype similar to AI [9]. Through alternate splicing, at least 15 mRNAs are generated in the ameloblasts from a single amelogenin gene [10–12], which consists of 9 exons [11, 13] nested in an intron of ARHGAP6 gene located in chromosome X [14]. M180, the major amelogenin isoform, is the product of exons 2,3,5,6 and 7 [15]. The largest exon, exon 6, contains three internal splice acceptor sites. The major cryptic splice deletion removes most of the 5-prime end of exon 6, resulting in the isoform leucine-rich amelogenin peptide (LRAP), also known as M59 [15]. *In vivo* and *in vitro* experiments confirmed an important role of amelogenins in enamel formation, especially M180 in enamel thickness and crystal elongations of hydroxyapatite [16–18]. However, functional significance of other splice variants including LRAP is not clearly understood.

ECM can provide not only the scaffolds for cellular processes, but also provide critical signals that can regulate the cell type or tissue specific cell growth, migration, differentiation and apoptosis [19]. For example, during tooth development, dentin phosphoprotein (also called phosphophoryn), one of the major ECM proteins in tooth dentin, drives cell differentiation of both the human mesenchymal stromal cells (MSCs) and MC3T3-E1 osteoblastic cell line through integrin signaling and activation of the MAPK pathway [20], and of CH310T1/2 undifferentiated mouse cells through integrin signaling by activation of focal adhesion complexes [21]. Dentin matrix protein 1 (DMP1), another major dentin ECM, not only supports the initiation of hydroxyapatite nucleation at the dentin mineralization front as a matrix protein, but also acts as a potential transcriptional regulator with respect to the regulation of other dentin proteins [22]. Recently, enamel-related matrix proteins including the amelogenins and ameloblastin have been detected not only in the ameloblasts, which are differentiated from an epithelial cell lineage, but also in other tissues including mesenchymal tissues at low level [11, 23, 24], suggesting enamel-related matrix proteins might possess other distinct functions in these tissues. In particular, the amelogenins have been also implicated in tissue-specific epithelial-mesenchymal signaling [15, 25–28] in addition to amelogenins' role in enamel formation as ECMs.

Therapeutic applications of an enamel matrix derivative (Emdogain®, Biora AB, Malmö, Sweden) rich in amelogenins have been reported to result in regeneration of cementum, surrounding alveolar bone and periodontal ligaments (PDLs) in the experimental treatment of periodontitis or in the treatments for human periodontitis [29–32], indicating the attractive

potential of amelogenin for hard tissue formation and regeneration [33]. Interestingly, bone sialoprotein (*Bsp*) mRNA expression in amelogenin null mice is significantly reduced not only in cementoblasts and surrounding osteoblasts [27] but also in calvarial bones [24]. Earlier *in vitro* studies suggested that the recombinant M180 and LRAP purified from bacteria have abilities to control the gene expression in an osteoblastic cell line [34] and cementoblastic cell line [27], respectively. By utilizing the chemically synthesized LRAP, a previous report suggested that osteogenesis in mouse embryonic stem cells was induced by activation of the Wnt pathway [35]. Moreover it has been reported that the amelogenin induced osteogenesis occurred when LRAP was injected in muscle tissues [26]. The past literature indicated that, among the many splicing isoforms, LRAP may play a pivotal role in inducing osteogenesis. Beside osteogenesis, LRAP has also significantly inhibited the osteoclastogenesis *in vitro* compared with M180 based on our previous report [36]. Therefore, we hypothesized that the amelogenins, especially LRAP, might function as a signaling molecule in bone metabolism.

The present study was undertaken to investigate the *in vivo* functions of LRAP in bone metabolism. Here we show that the LRAP transgenic (TgLRAP) mice resisted the bone loss induced by ovariectomy. LRAP brings about these effects possibly through inhibition of RANKL–mediated osteoclastogenic pathway and by accelerating bone formation. The results from this study identify not only the novel functions of enamel matrix proteins, but also shed light on the new mechanisms, regulating systemic bone homeostasis *in vivo*.

## Materials and methods

### Generation of LRAP transgenic mice and genotyping

An *Asp 718-BamHI* 2.3-kb fragment of the mouse collagen alpha 1 type I promoter (Col1a1) was released by restriction endonuclease digestion of pJ251 (courtesy gift from Dr. de Crombrugghe) [37] and a *BamHI-SspI* 1.6-kb fragment containing IRES (internal ribosome entry site)-EGFP (enhanced green fluorescent protein)-SV40 polyA released by restriction endonuclease digestion of pIRES2-EGFP (Clontech) were simultaneously cloned into the *KpnI* and *EcoRV* sites of pcDNA3.1(+) (Thermo Fisher Scientific). The mouse LRAP cDNA was amplified from C57B6 tooth cDNA and cloned in a pEF6-V5His TOPO vector (Thermo Fisher Scientific) to segregate from the other amelogenin isoforms and to verify the LRAP sequence. The LRAP cDNA was again amplified by polymerase chain reaction (PCR) using KOD HS DNA Polymerase (Novagen) and primers with *BamHI* adapter sequences (5′–CG<u>GGATCC</u>GCCA TGGGGACCTGGATTTTG–3′ and 5′–CG<u>GGATCC</u>TTAATCCACTTCTTCCCGCTTG–3′). The amplified fragment was digested by *BamHI* and introduced in between the collagen promoter and IRES-EGFP-SV40 polyA. The orientation and accuracy of the LRAP cDNA were checked by DNA sequencing. The whole transgene DNA construct (about 4.1 kb) was removed at the *NheI* and *XhoI* sites from the pcDNA3.1(+) vector backbone (Fig 1A).

Transgenic mice were generated by microinjection of the DNA construct into pronuclei from FVB/N mouse eggs [38]. Genomic tail DNA samples digested by *BamHI* were used for Southern blot hybridization using a Col1a1-LRAP-IRES-EGFP cDNA probe labeled with $P^{32}$ to identify mice positive for the presence of transgene. Three founder mice were identified, and two selected founders were further mated to FVB/N mouse strain to establish the independent transgenic lines. The genotype of the F1 or later generations was screened by checking the EGFP expression in teeth with the Illumatool Bright Light System (LT-9900, Lightools Research) in a dark box, and then confirmed by PCR amplification of genomic tail DNA using primers in S1 Table. Hemizygote transgenic mice (TgLRAP) overexpressing LRAP were compared to their wild-type (WT) littermates between F2 to F4 generations. Female mice were used for the following analysis except for the mice at 3- and 5-day-old. All mice were housed

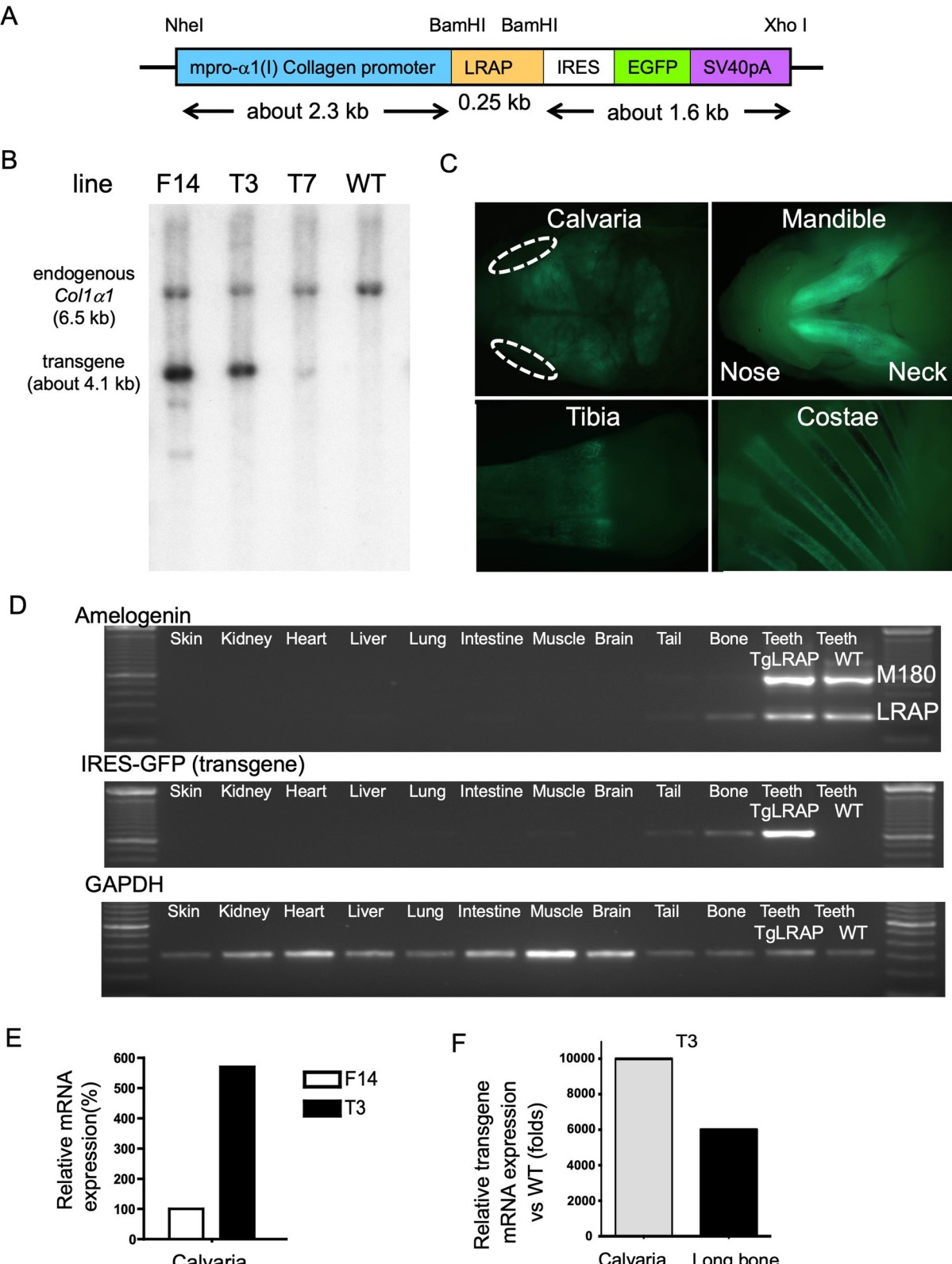

**Fig 1. Generation of LRAP transgenic (TgLRAP) mice and transgene expression profiling. A**, Construct for LRAP transgenic mouse. Mouse LRAP cDNA and an enhanced green fluorescence protein (EGFP) coding sequence with IRES sequence were tandemly ligated under the control of murine 2.3kb type I collagen promoter. The DNA fragment from *NheI* to *XhoI* was microinjected to pronuclei of mouse eggs.

The two independent proteins, LRAP and EGFP were translated from a single mRNA. **B**, Southern blotting for genotyping the founder mice. Tail DNA samples were digested with *BamHI*. The entire transgene from *NheI* and *XhoI* depicted in panel A was used as a radiolabeled probe. The endogenous collagen promoter and transgene sequence appeared as a band around 6.5 and 4.1 kb, respectively. Based on the calculation of band intensities, line F14, T3 and T7 appeared to carry 3, 2, and 0.6 copies of transgene, respectively. This figure panel 1B is reproduced from our earlier publication (Haruyama N, Cho A, and Kulkarni AB. 2009. Overview: Engineering Transgenic Constructs and Mice. Current Protocols in Cell Biology 19.10.1–19.10.9. DOI:10.1002/0471143030.cb1910s42. All Copyrights reserved as per Creative Commons license). **C**, EGFP expression visualized under epifluorescence. Bone tissues and teeth (dentin) showed bright fluorescence in 3-day-old transgenic mice. Dotted lines; eyes. **D**, Screening for transgene expression in major organs. RT-PCR was performed to detect transgene expression in various tissues from transgenic mice (TgLRAP) at 5-day-old. The transgene expression was predominantly detected in teeth, bone, and tail tissue. WT teeth act as positive and negative controls from the littermates for the expression of amelogenin and IRES-EGFP (transgene), respectively. GAPDH; glyceraldehyde 3-phosphate dehydrogenase as an internal control. **E**, Relative mRNA expression levels of transgene in calvaria of line F14 and T3 mice at 5-day-old. The empty and solid columns represent the transgene expression levels in line F14 and T3, respectively. The level of F14 is depicted as 100%. Although line T3 harbored a lower copy number compared to line F14, the relative mRNA expression level in calvaria was ~6 times higher than that in line F14, possibly due to the difference of integration locus of the transgene. **F**, Relative LRAP mRNA expression level in line T3 compared to the endogenous level in WT. 5-day-old TgLRAP calvaria expressed 10,000-fold more LRAP than WT calvaria; the expression was around 6,000-fold more compared to the endogenous level in long bone. WT; wild-type mouse, TgLRAP; LRAP transgenic mouse.

in a specific pathogen-free animal facility and fed food and autoclaved water *ad libitum*. All animal studies were conducted in compliance with the NIH guidelines for the Care and Use of Laboratory Animals and approved by the Institutional Animal Care and Use Committee (IACUC) of the National Institute of Dental and Craniofacial Research.

EGFP expression in 3-day-old TgLRAP mice was visualized by a stereomicroscope under epifluorescence (Stemi2000, Zeiss) to check the expression pattern of transgene visually. Major organs from a 3-day-old TgLRAP and WT littermate mice were dissected, and total RNA was isolated from the organs using Trizol reagents (Thermo Fisher Scientific) followed by DNase I (Promega) treatments. The RNA samples (0.5 μg each) were subjected to first strand cDNA synthesis using SuperScript II First-strand Synthesis System (Thermo Fisher Scientific). All RT-PCR reactions were carried out in a Mastercycler Epgradient S PCR machine (Eppendorf). Quantitative PCR (Q-PCR) analysis was also performed to investigate the transgene expression levels in calvarial bones and hind limb long bones. Total RNA was isolated from 5-day-old mice as described above, followed by RNeasy purification with DNase I on-column treatments (Qiagen). The RNA samples (2 μg each) were subjected to first strand cDNA synthesis using the SuperScript III first-strand synthesis system (Thermo Fisher Scientific). All Q-PCR reactions were prepared with iQ SYBR Green supermix and carried out in a PTC-200 thermal cycler with Chromo4 Real-Time PCR Detection System (Bio-Rad). RT-PCR or Q-PCR was performed using primer sets described in S1 Table.

### *In vitro* analysis of osteoclastogenesis

For the direct induction of osteoclastogenesis from spleen, splenocytes were collected aseptically from spleens of 1-month-old mice. After the removal of red blood cells with ammonium–chloride–potassium (ACK) lysing buffer (Cambrex), the cells were cultured overnight (20 x $10^6$ cells/10cm dish) at 37°C and 5% $CO_2$ in MEM-α (Gibco, Thermo Fisher Scientific) containing 10% fetal bovine serum (FBS) (HyClone), supplemented with M-CSF (50 ng/ml) (R&D systems). Non-adherent cells were resuspended in culture medium and plated at $2 \times 10^5$ cells/well into 96-well plates supplemented with M-CSF (50 ng/ml) and RANKL (100 ng/ml) (R&D systems). The medium was changed on days 3, 5, and 7 to the culture medium containing fresh M-CSF and RANKL. The osteoclastogenesis was evaluated by TRAP staining kit (Sigma) at 9 days of culture.

For the osteoclastogenesis in the coculture system, primary osteoblasts were isolated from calvaria of 1- to 2-day-old mice. Calvarial bones were sequentially digested for 10 min in Hanks' balanced salt solution containing 0.1% collagenase class 1 (Worthington) and 0.2%

dispase II (Roche diagnostics). The cells isolated from fractions 2 to 5 were combined as osteo-blastic cell populations and expanded in 10 cm dishes in MEM-α containing 10% FBS. After reaching 80% confluency (i.e., after 6 to 8 days), the calvarial osteoblasts ($2 \times 10^4$ cells/well) were plated in 24-well plates the day before starting the co-culture with ACK treated fresh bone marrow cells (BMCs) ($1 \times 10^6$ cells/well) for osteoclastogenesis. The osteoclastogenesis was evaluated by TRAP staining kit (Sigma) at 7 days of culture in MEM-α supplemented with 50 mg/L ascorbic acid.

## Osteoinduction of primary calvarial osteoblasts from TgLRAP mice and analysis for osteoblastic cell differentiation

Primary osteoblasts were isolated as described above and plated in 35-mm dishes to investigate the alkaline phosphatase (ALP) activity and calcium deposition. After 10 days of culture in MEM-α supplemented with 50 mg/L ascorbic acid and 5mM β-glycerophosphate (Sigma), half of the calvarial osteoblasts were rinsed with 0.01 M phosphate buffered saline (PBS), and then fixed with 4% paraformaldehyde (PFA) in PBS. The ALP activity was visualized by blue stain-ing with fast blue BB salt solution (Fluka) including naphthol AS-MX phosphate (Sigma) as a substrate. The ALP activity of the cell lysate was also quantified by colorimetric assay (LabAs-say ALP, Wako USA). To normalize the ALP activity, fractions of the cell lysate were used to examine the total protein concentration by BCA protein assay kit (Pierce). To compare the abilities for calcium deposition, the other half of calvarial osteoblasts were stained with alizarin red S (Sigma) at 21 days of culture. Calcium accumulation was quantified by the measurement of acid-soluble calcium with the o-cresolphthalein complexone kit (Calcium C-test kit, Wako).

Q-PCR analyses of the gene expressions in the cultured calvarial osteoblasts were per-formed at 10 days of culture in the presence of ascorbic acid (50μM). The cDNA synthesis and reactions for Q-PCR were prepared as described above. The results were normalized to the expression level of HPRT as an internal control. Expression of genes, listed in S1 Table, involved in bone formation *(Runx2, Alp, Bsp, Oc,* and *Opn)* or bone resorption *(Mmp13, Rankl,* and *Opg)* were analyzed by $2^{-\Delta/\Delta C_T}$ method [39]. The primer sequences for Q-PCR were acquired from Primer Bank (http://pga.mgh.harvard.edu/primerbank/) [40]. Q-PCR products were subcloned into a pCRII TOPO vector (Thermo Fisher Scientific), and the amplicon's nucleotide sequences were confirmed by cycle sequencing.

## Whole mount skeletal staining, tissue sections preparation, immunohistochemistry, and TRAP staining

For whole mount skeletal staining, the 5-day-old WT and TgLRAP mice were double stained with alcian blue 8GX (Sigma) and alizarin red S (Sigma) for 10 days. In addition, the 1-month-old TgLRAP mice and the littermate WT controls were used for the histological analysis. Six mice in each group were anesthetized and perfused with 4% PFA in 0.1 M PBS, pH 7.4. After dissection, the tibiae were fixed in 4% PFA for 24 hours, decalcified in 10% ethylenediamine-tetraacetic acid (EDTA) in 0.01 M PBS, pH 7.4, for 4 to 6 weeks at 4˚C, dehydrated in a graded series of ethanol, embedded in paraffin, and serially sectioned at 7-μm thickness. To localize the transgene expression, immunohistochemistry for EGFP was performed. The sections were blocked with 10% normal goat serum (Thermo Fisher Scientific) and incubated with rabbit polyclonal antibody against GFP (Thermo Fisher Scientific) at 1:1000 dilutions in PBS. The peroxidase reaction in the immune complexes was visualized by a chromogen substrate 3-amino-9-ethyl carbazole (AEC) reaction according to the manufacturer's instructions (SuperPicTure Polymer Detection Kit, Thermo Fisher Scientific). The nuclear counterstaining was performed using Hematoxylin for 5 seconds. To examine the co-expression of LRAP and

EGFP in the tissue sections, immunofluorescence was performed. The primary antibodies for the GFP or the rabbit polyclonal anti amelogenin carboxy-terminus [41] were labeled with Alexa Fluor 488 or Alexa Fluor 594, respectively (Zenon Tricolor Rabbit IgG Labeling Kit #2, Thermo Fisher Scientific). After overnight incubation with the antibodies at 4˚C, the nuclear staining was performed with DAPI (Prolong, Thermo Fisher Scientific). For the analysis of osteoclastogenesis, ten sections from each 1-month-old male tibia were stained to detect tartrate-resistant acid phosphatase (TRAP) activity with the Leukocyte Acid Phosphatase kit (Sigma) and counterstained with toluidine blue. The number of osteoclasts per bone surface perimeter (N.Oc/B.Pm) were determined by Image J (http://rsbweb.nih.gov/ij/).

### Tests for bone turnover markers, and osteoclast histomorphometry in sham-operated (sham) or ovariectomized (OVX) mice

For sham operation or ovariectomy in WT and TgLRAP mice, four 8-week-old female mice from each group were anesthetized by isoflurane (2.5–4.5% in $O_2$) prior to the surgery. The mice were euthanized at 4 weeks post-surgery for the following radiographic and histological assays. Peripheral blood was harvested by intraorbital approach from the mice at 4 weeks post-surgery of Sham/OVX. Cytokines and bone resorption markers including mouse IL-1β (Biosource, CA), RANKL and OPG (R&D systems) in serum were measured by enzyme-linked immunosorbent assay (ELISA). The right femora samples from Sham/OVX mouse were prepared for tissue sections as described above. Ten sections from each femur were investigated for the analysis of osteoclasts. Osteoclasts were identified as TRAP (+) multinucleated cells adjacent to trabecular bone. The number of osteoclasts per bone surface perimeter (N.Oc/B. Pm) were determined by Image J. All measurements were confined to the secondary spongiosa and restricted to the trabecular bone in an area between 0.5 to 1 mm distal from the bone growth plate-metaphyseal junction. The left femora were maintained in PBS after the PFA fixation for further radiographic analysis.

### Radiographic and micro computed tomography (µCT) analysis

WT and TgLRAP femora from 12-week-old Sham/OVX mice were analyzed with a Faxitron MS-20 specimen radiography system (Faxitron X-ray) for 90 s at 18 kV using diagnostic films (X-OMAT, Kodak). The femora were further used for three-dimensional (3D) micro-computed tomographic (µCT) analyses (eXplore MS, GE Medical Systems). The femora and a calibration phantom (SB3, Gammex RMI) were scanned with 8-µm isotropic voxels (80 kV, 80 µA) on the µCT system and the reconstructed 3D images of distal femora were analyzed using Micro-View software (GE Medical Systems). The trabecular (cancellous) bone volume/ tissue volume ratio (BV/TV; %) and the trabecular bone mineral density (BMD; mgHA/cm$^3$) as hydroxyapatite (HA) equivalents in a cylindrical area (ROI; Ø1.0 x 0.5 mm) (Fig 5B) within the distal metaphyseal area underneath the growth plate-metaphyseal junction were analyzed [42].

### Quantitative analysis of bone formation

Six 1-month-old male and four Sham/OVX operated female mice received fluorochrome labels by intraperitoneal injection for evaluation of bone dynamics (2 and 7 days prior to sacrifice, 20 mg of calcein (Sigma) in 2% NaHCO$_3$/kg of body weight). The heads were removed and fixed in 70% EtOH overnight at room temperature in the dark to eliminate the EGFP fluorescence activity. The calvarial bones were visualized under UV light and recorded digitally using the AlphaImager system (Alpha Innotech). The digital images were then analyzed using Image J software to quantify the calcein intakes [43]. For presentation, the images were pseudocolored in Image J using the green setting.

### Evaluation and statistical analysis

The results are expressed as means ± standard deviations (SD). Statistical analysis was carried out using the student t-test or ANOVA followed by Tukey-Kramer test (Prism6, GraphPad software). All statistical tests were unpaired comparisons. $P<0.05$ was considered statistically significant.

## Results

### Generation of LRAP transgenic (TgLRAP) mice and the transgene expression

To determine whether LRAP has a role in bone metabolism, we generated a transgenic mouse where we express LRAP along with and IRES-EGFP under the control of a Col1a1 promoter. Based on the genotyping results by Southern blotting (Fig 1B), we initially obtained three TgLRAP founders named F14, T3 and T7. The line T3 harbored a lower copy number (~2 copies) of transgenes as compared to line F14 (~3 copies) based on the band intensities of Southern blotting. Lines F14 and T3 were established as independent lines and used for further analysis. Transgene expression patterns of the two lines were checked on the 3-day-old F1 mice by epifluorescence, revealing that the two lines displayed the collagen promoter specific-expression pattern of EGFP, predominantly in the teeth and the bones such as calvaria, mandible, tibia, and costae (Fig 1C), as previously reported [37]. To investigate precise transgene expression patterns in the transgenic mice, RT-PCR for each major organ was performed (Fig 1D). The results indicated that there was no leaky transgene expression: *i.e.*, the transgene was predominantly expressed in bone and teeth (dentin). The transgene expression level and patterns were further investigated in bones by Q-PCR. Although the line T3 harbored lower copy number compared to the line F14 (Fig 1B), the relative mRNA expression levels in bones were more than 3–6 times higher than that in the line F14 at 5-day-old (Fig 1E), which possibly was due to the difference of integration locus of the transgene. The LRAP expression level in the calvaria or long bone of line T3 TgLRAP mice reached 10000 or 6000 times higher than that in the WT mice, respectively (Fig 1F).

Protein expressions of EGFP and LRAP by the transgene were confirmed in bone sections by immunohistochemistry. Based on the staining pattern, most of the signal from transgene expression was detected in osteoblasts and osteocyte-like cells located in the bone matrix in the bone sections (Fig 2B, red arrow heads). EGFP (green) and LRAP (red) were co-expressed in the same cells (Fig 2D, yellow cells indicated by white arrows) in TgLRAP sections, although faint red signals from endogenous amelogenin were detected in osteoblasts from both WT and TgLRAP mice sections (Fig 2C and 2D; red arrows). Several cells, which were morphologically like osteocytes, showed the signal only for EGFP (Fig 2D, white arrow heads), suggesting the slower turnover of EGFP in those cells.

The TgLRAP mice were normal at birth and show no apparent defects during the first 3 weeks of life. The TgLRAP mice did not present growth retardation or an increased incidence of lethality, either. All the *in vivo* and *in vitro* data shown is from line T3, but was also confirmed in line F14, which showed almost identical phenotypes and the expression patterns of transgene.

### TgLRAP mice showed increased osteoblastic activities and decreased osteoclastogenesis *in vitro*

The calvarial osteoblasts and/or BMCs from WT or TgLRAP mice were isolated and tested *in vitro*. The cultured osteoblast cells from TgLRAP mice showed more intense staining for ALP

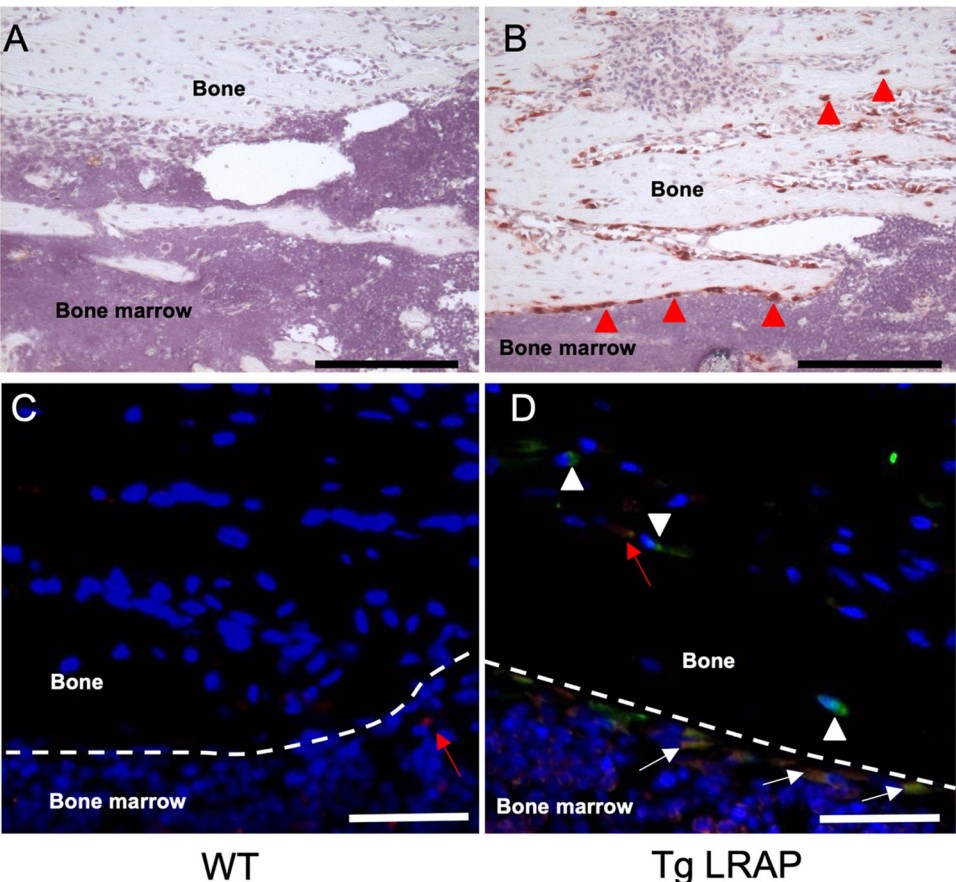

**Fig 2. Expression pattern of the transgenes by immunohistochemistry.** Transgene expression examined by immunohistochemistry using 1-month-old tibia tissue sections. **A**, No background signal was detected by anti-GFP antibody in wild-type mice. **B**, Intense signal from the anti-GFP antibody was detected in both the osteoblasts and osteocyte-like cells (red arrow heads) located in the bone matrix. **C-D,** Immunofluorescence staining to examine co-localization of LRAP and EGFP in tibia sections. Very faint signals were detected from WT tissue by the anti-amelogenin antibody (**C**, red arrow). EGFP (green) and LRAP (red) were co-expressed in most of the osteoblast like cells in TgLRAP (**D**, yellow cells indicated by white arrows). Cells that appeared to be osteocytes showed only the EGFP signal, suggesting the LRAP might be quickly degraded compared to EGFP in some cells (white arrow heads). On the other hand, endogenous amelogenin was detected in osteocyte-like cells located in the bone matrix without EGFP signal (red arrow). Red; anti-amelogenin antibody, Green; anti-GFP antibody, Yellow; merged. Dashed lines: border between bone matrix and bone marrow, Black bars = 200 μm, White bars = 50 μm. WT; wild-type mouse, TgLRAP; LRAP transgenic mouse.

activity than that in the WT calvarial cells at 10 days of culture (Fig 3A). In addition, alizarin red staining demonstrated that *in vitro* mineralization of TgLRAP calvarial cells was increased compared with the WT calvarial cells at 21 days of culture (Fig 3B). Both relative ALP activities and accumulated calcium amount were statistically greater in TgLRAP cells compared to WT (Fig 3A and 3B, graphs). These results suggested that the LRAP promoted the differentiation of osteoblastic cells. On the other hand, direct induction of osteoclastogenesis with RANKL/M-CSF from spleen cells did not show any significant difference between WT and TgLRAP cells (Fig 3C). The co-cultures of calvarial cells and BMCs were further performed to investigate the effect of LRAP on osteoclastogenesis in the presence of large population of mesenchymal cells. The co-cultures demonstrated that the largest number of TRAP (+) osteoclasts was formed when the WT BMCs were co-cultured with WT calvarial cells (Fig 3D; WT / WT), whereas the fewer number of osteoclasts was formed when the TgLRAP BMCs were cultured

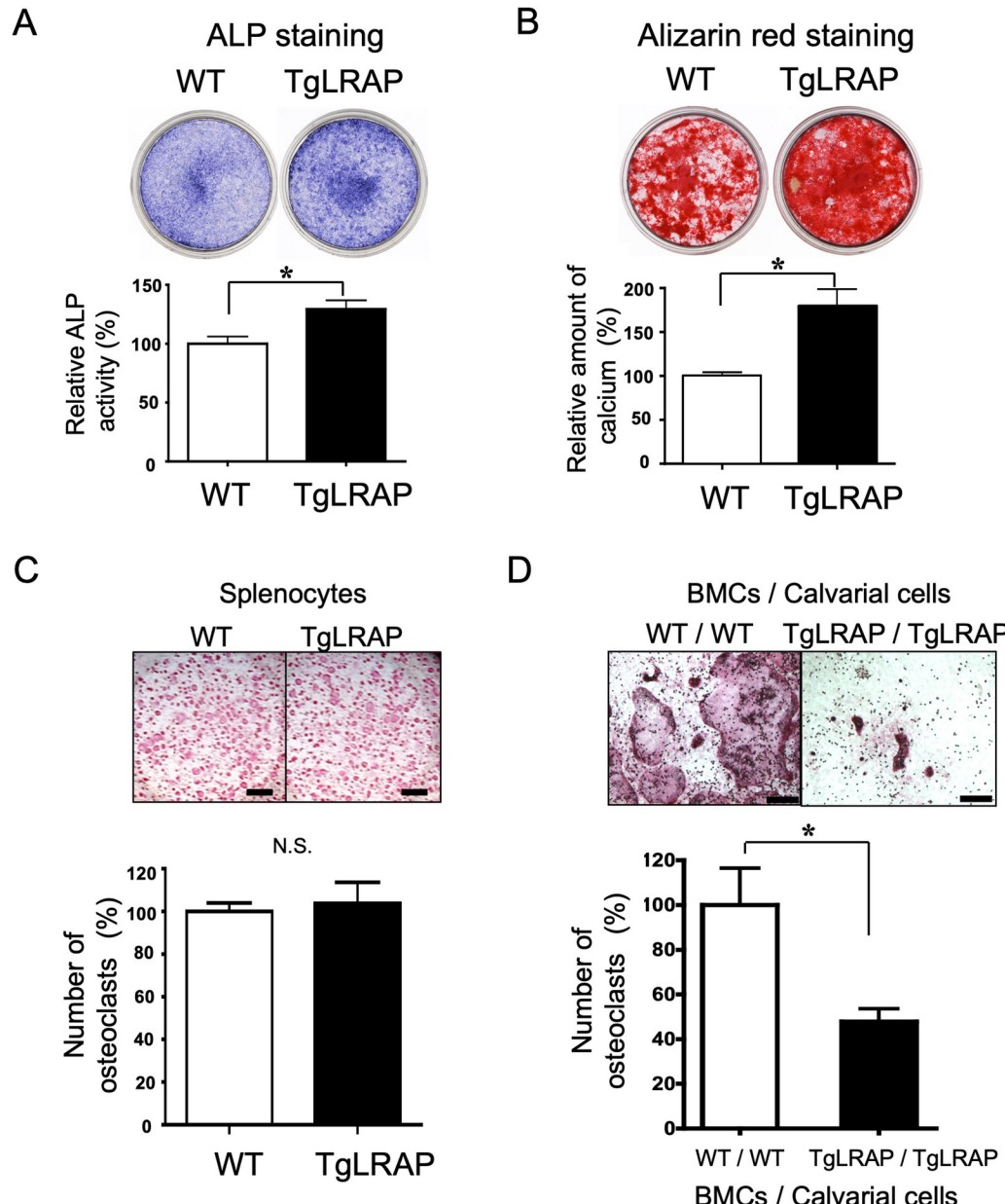

**Fig 3. Increased osteoblastic activity and decreased osteoclastogenesis in TgLRAP cells *in vitro*. A and B,**
Representative examples of ALP and alizarin red staining of osteoblastic cells isolated from calvaria bones. The
experiments were repeated 3 times. Relative ALP activities (**A**, at day 10) and accumulated calcium amount (**B**, at day 21)
were statistically higher in the TgLRAP cells as compared to the WT (graphs). **C,** TRAP staining of spleen cells cultured
with RANKL and M-CSF. The numbers of osteoclasts (TRAP positive cells) were compared at 9 days of culture. No
differences were found between WT and TgLRAP. Bars = 1 mm. **D,** TRAP staining of co-cultured BMCs and calvarial
cells at 7 days of osteoclast induction. Compared with the co-culture of WT BMCs and WT calvarial cells (WT / WT), the
number of osteoclasts were significantly decreased at 50% in the co-cultures of TgLRAP BMCs and TgLRAP calvarial cells
(TgLRAP / TgLRAP) (graph). Bars = 500 μm. The experiments were repeated 3 times. Each column represents the
mean ± SD. *p<0.05. WT; wild-type mouse, TgLRAP; LRAP transgenic mouse.

with TgLRAP calvarial cells (Fig 3D; TgLRAP / TgLRAP). These results suggested the inhibi-
tory effects of calvarial and bone marrow mesenchymal cells obtained from TgLRAP mice on
the osteoclastogenesis *in vitro*.

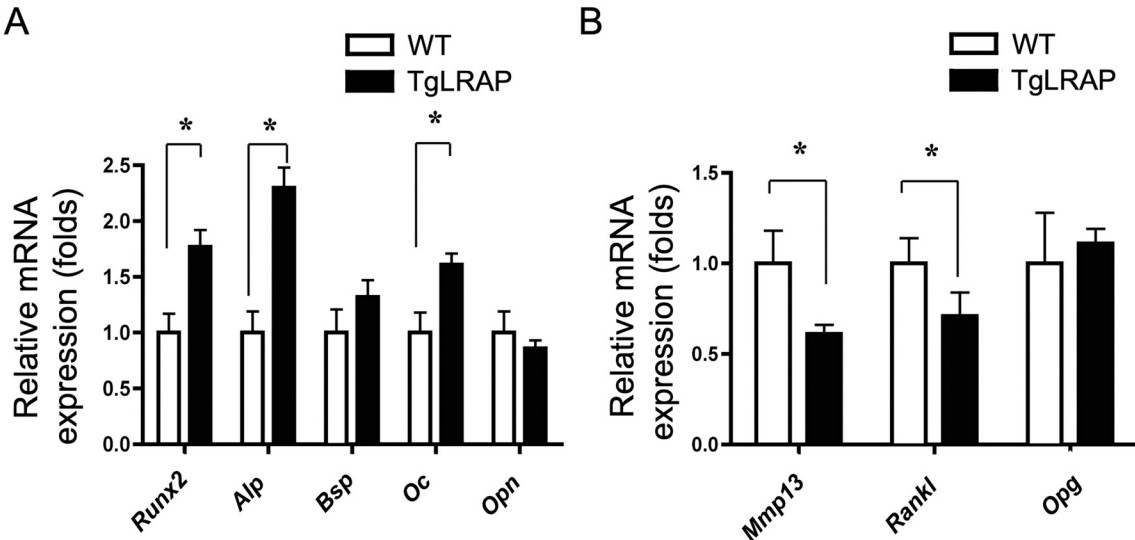

**Fig 4. Altered gene expression levels in LRAP transgenic calvarial cells using Q-PCR.** The empty or solid column represents the mRNA expression of WT or TgLRAP calvarial cells examined, respectively. **A,** Bone formation markers. **B,** Bone resorption makers. The experiments were repeated 3 times. Each column represents the mean ± SD. $^*p<0.05$. WT; wild-type mouse, TgLRAP; LRAP transgenic mouse.

To understand how the LRAP affected the promotion of osteoblastic differentiation and the inhibition of osteoclastogenesis, we investigated mRNA expression levels of the genes known to be involved in bone formation and bone resorption in calvarial cells (Fig 4A and 4B, respectively). TgLRAP calvarial cells showed higher *Runx2*, *Alp* and *Oc* expression compared to WT. Meanwhile, *Mmp13* and *Rankl* expression was decreased compared to WT, although expression of osteoprotegerin (*Opg*), bone sialoprotein, and osteopontin were not altered. These results suggested that the LRAP had the effects on calvarial cells, changing their character to more anabolic (osteogenic) *in vitro*.

## TgLRAP mice resisted bone loss induced by OVX

We analyzed both the calvaria and hind limb long bones since we wanted to examine the phenotypic and expressional differences between bones that ossify with membranous ossification or endochondral ossification. Some mouse models showed different phenotypic changes in each bone with different ossification styles according to the expression levels. For example, heterozygous *Cbfa1*$^{+/-}$ embryos exhibited phenotypic changes predominantly in clavicles and calvarial bones [44]. Here, we identified that the amelogenin expression levels in the calvarial bone and the hind limb long bones were different (Fig 1F). However, there were no obvious phenotypic changes or differences in the calvarial and hind limb bones of TgLRAP (S1 Fig). Phenotypic changes in bone samples from WT and TgLRAP were also compared at different ages. Neither calvaria formations in 5-day-old mice (S1A Fig) nor calcein uptake into calvaria of 1-month-old male mice showed significant differences, although the fluorescence intensity was higher in TgLRAP mice (S1B Fig). The number of osteoclasts in tibia indicated that there are no significant changes in bone resorption in 1-month old TgLRAP mice as compared to the controls (S1C Fig).

To investigate the role of LRAP under the situation with rapid bone turnover, we performed OVX in female mice, leading to experimental bone loss as previously reported [45]. The OVX WT mice showed bone loss as compared to other groups by fine focus radiographic analysis; the OVX TgLRAP mice did not show significant differences as compared to Sham WT or TgLRAP mice (Fig 5A). The µCT analysis of ROI (Fig 5B) in the femur revealed that

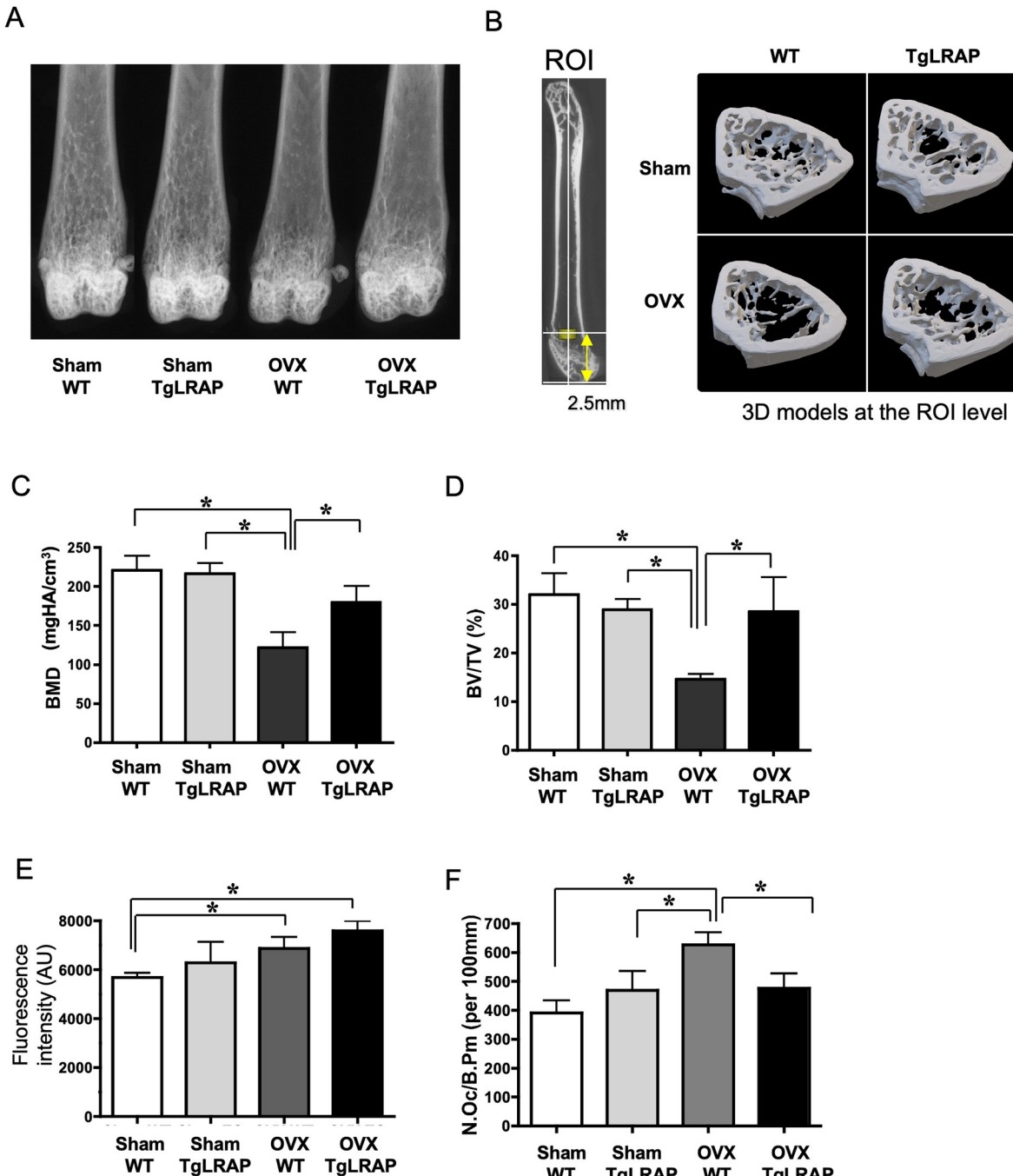

**Fig 5. TgLRAP mice resisted bone loss induced by ovariectomy (OVX). A,** Fine focused radiographic analysis of femur. OVX WT showed decreased trabecular bone structures compared to other groups. **B,** 3D μCT evaluations of bone parameters. The left panel indicates the region of interest (ROI; yellow cylinder) for following measurements. The right panel indicates the tomography at the level of ROI. **C and D,** Parameters analyzed by μCT. BMD and BV/TV were decreased in OVX WT compared to other groups. BMD; bone mineral density. BV/TV; ratio of trabecular bone volume per total bone volume, each column represents the mean ± SD. n = 4 for each group. **E,** Quantitation of calcein incorporation into the calvaria. AU; arbitrary unit. Each column represents the mean ± SD. n = 4 for each group. **F,** Histological quantitation of bone resorption in sham-operated (Sham) /OVX WT or TgLRAP mice. The number of osteoclasts per bone perimeter (N. Oc/B.Pm (per 100 mm)) were compared. Each column represents the mean ± SD. n = 4 for each group. *p<0.05. WT; wild-type mouse, TgLRAP; LRAP transgenic mouse.

the BMD and BV/TV were significantly decreased in OVX WT as compared to other groups (Fig 5C and 5D). These results suggest that the TgLRAP mice resisted the bone loss induced by OVX. The quantitative analysis of bone formation in 12-week-old Sham/OVX mice revealed that the OVX WT and TgLRAP mice had increased bone formation (increased calcein uptake) in calvaria compared to Sham WT mice (Figs 5E and S2A). Although the OVX resulted in increased bone formation in both WT and TgLRAP mice, OVX TgLRAP appeared to show the most remarkably increased bone formation. The parameters for bone resorption checked by TRAP staining showed increased number of osteoclasts (N.OC/B.Pm) in OVX WT, but not in OVX TgLRAP (Figs 5F and S2B) compared to Sham WT or Sham TgLRAP mice. To understand the reason for suppressed osteoclast formation, serum cytokine levels including IL-1β, RANKL and OPG, those are closely related to osteoclastogenesis, were measured by ELISA. However, none of those cytokine levels showed significant differences between the experimental groups (S3A–S3C Fig).

## Discussion

Here we report that the TgLRAP mice overexpressing LRAP under the control of type I collagen promoter showed altered bone remodeling when the mice were challenged by means of OVX. Although *in vivo* expression of LRAP did not affect main bone characteristics, the OVX experiments demonstrate that the OVX TgLRAP mice resist induced bone loss by OVX, whereas OVX WT mice suffer bone loss as compared to sham-operated mice. This is the first evidence that the LRAP, one of the amelogenin splice variants, is implicated in systemic bone turnover *in vivo* with a specific condition such as higher bone turn over.

In this study, we have generated two independent lines of transgenic mice. Although each line carried different copy numbers of transgene and expressed different amount of transgene products (Fig 1), the phenotypic analysis revealed that the two lines represented very similar phenotypes both *in vivo* and *in vitro*. From the expression pattern analysis of transgene, the expression has been successfully induced predominantly in osteoblasts and partially osteocyte-like cells in this study (Figs 1 and 2). The 2.3kb type I collagen promoter could strongly and faithfully drive the transgene with similar expression pattern to the previous report [37, 46]. Our previous report demonstrated that the loss of amelogenin expression in amelogenin KO mice resulted in altered calvarial bone development and size *in vivo* and reduced mineralization of calvarial osteoblasts from amelogenin KO mice in vitro [24]. Since we planned to investigate the *in vivo* functions of LRAP in bone metabolism, we believe that the 2.3kb type I collagen promoter activity should suit our study and that LRAP expressed by osteoblasts have affected the bone phenotypes.

Actually, the detection level of LRAP on the TgLRAP tissue sections appeared to be lower than the expected level based on the numerous transcript number of TgLRAP (Fig 2). The reason for the low detection level of LRAP by antibody in bone would be that the secreted protein is generally detected at a relatively lower level with immunohistochemistry as compared to other proteins confined in the cytoplasm. We do not know the physiologic or minimal level for LRAP to induce bone formation or to inhibit osteoclastogenesis. However, the endogenous expression level of amelogenin is enough to change the osteoblastic activity of ES cells [35] and calvarial osteoblasts [24], or the osteoclastogenesis by PDL cells [28]. Therefore, we believe that the amount of LRAP should be above the biologically effective dose since the expression level of amelogenin protein in the TgLRAP bone tissues was higher than the WT control.

To investigate the functions of LRAP, first we performed *in vitro* experiments and confirmed the increased bone formation and decreased bone resorption in the cells derived from TgLRAP mice (Fig 3). Notable results from the *in vitro* experiments are the LRAP effects on

both osteogenesis and ability to support osteoclastogenesis of mesenchymal cells, that express LRAP by themselves, through possibly regulating the gene expression of bone formation and osteoclastogenic markers (Fig 4). Increased expression of *Runx2*, *Alp*, and *Oc* in TgLRAP cells was consistent with the results from ALP and alizarin red staining. Interestingly, BSP expression was not altered in this culture system, although previous *in vitro* [27, 34] and *in vivo* [24, 27] reports demonstrated that the BSP expression was affected by the presence of M180 or the absence of amelogenins, respectively. Other previous reports suggested that exogenous addition of LRAP induces the BSP expression and osteogenesis in mouse embryonic stem cells, possibly through the activation of the canonical Wnt signaling pathway [35, 47]. We speculate that the changes of BSP shown in past reports might be specific to the M180 isoform or to the embryonic stem cells. In addition to the increased expression of bone formation markers, the Q-PCR results indicated that the RANKL/OPG ratio was altered by LRAP by suppression of RANKL expression in TgLRAP cells. The synergistic inhibition of osteoclastogenesis observed in co-cultures might be from the effects of LRAP on the inhibition of RANKL expression through mesenchymal cells.

Despite the abundant expression of transgene and the results from *in vitro*, significant phenotypic changes were not found in TgLRAP mice either at 5-day-old or at 1-month-old (S1 Fig). We have already demonstrated that LRAP is the negative regulator of osteoclastogenesis in co-cultures of osteoclast progenitor and cementoblast/periodontal ligament cells [36]. Experimental treatment using enamel derived matrix protein preparation (Emdogain), which contains a large portion of amelogenins, has been shown to be especially effective in treating bone area undergoing rapid turnover as found in periodontitis [48]. To reproduce experimental bone loss with rapid bone turnover and investigate the functions of LRAP, we performed ovariectomy (OVX) in female WT and TgLRAP mice (Fig 5). As a number of previous studies reported, OVX WT mice displayed significant bone loss indicated by several bone morphologic parameters compared to sham-operated mice. However, the OVX TgLRAP mice did not show significant changes in the bone parameters compared to Sham WT or Sham TgLRAP mice. The amount of bone formation indicated by calcein uptake was significantly increased, while the number of osteoclasts in bone tissues was not increased in OVX TgLRAP mice. These results suggest that the LRAP might have important roles in maintaining bone homeostasis by inducing bone formation and suppressing the osteoclast formation during the accelerated bone turn over.

The bone loss mechanisms underlying OVX or postmenopausal estrogen withdrawal have been well studied. Pacifici showed that monocytes isolated from post-menopausal women displayed increased secretion of interleukin-1β (IL-1β) and tumor necrosis factor- α (TNF- α), which could be reverted by estrogen replacement therapy [49]. These cytokines induce the expression of cyclooxygenase 2 (COX-2) in osteoblasts and BMCs, resulting in an increase in the production of prostaglandin $E_2$ ($PGE_2$) [50]. TNF-α, together with $PGE_2$, enhances the expression of RANKL, not only in stromal cells, but also in pre–B- lymphocytes, which then cooperatively stimulate osteoclast progenitors through RANK-RANKL interaction and promote the differentiation of these cells into mature osteoclasts [51]. To investigate in detail functions of systemic response to LRAP, the serum levels of cytokines related to bone metabolism were screened in Sham/OVX mice. However, IL-1β, RANKL or OPG showed no difference among all groups. The mechanisms how the bone phenotypes are regulated by LRAP should be further elucidated. However, it has been reported that the nitric oxide (NO) signaling pathway is activated in ameloblasts by adding exogenous LRAP *in vitro* [52]. NO derived from the eNOS pathway is known to act as a mediator of the effects of estrogen in bone [53]. Moreover, the effects of NO on bone marrow stromal cells are to decrease RANKL/OPG ratio [54]. Further studies defining the involvement of NO signaling pathway might help to clarify the mechanisms of action and activity of LRAP *in vivo*.

Recently, a transmembrane protein, LAMP-1, originally identified as a lysosomal membrane protein that is also found at the cell surface, has been found as an LRAP cell binding protein [55]. Mouse LAMP-3 (CD63) has been suggested as a binding partner for M180, but not for LRAP [56]. LAMPs are highly glycosylated trans-membrane proteins, associated with vesicular structures of the endocytic pathway. These proteins are especially implicated as a binding proteins functioning in scavenger process or protein uptake process after the degradation of amelogenin protein in tooth development (enamel formation) [57]. LAMP-3 (CD63) is expressed in osteoblastic cell lineage although the functional significance is still unknown [58]. Moreover, it is speculated that the functional difference between M180 and LRAP might be derived from the difference between ligand-receptor binding affinities among them. We have identified that the carcinoembryonic antigen (CEA)-related cell adhesion molecule 6 (CEA-CAM6, also known as CD66c), Leukocyte immunoglobulin-like receptor, subfamily B (LILR-B), and the integral membrane protein 2 (ITM2) family are the potential protein-binding partners for LRAP by a yeast two-hybrid assay [59]. These cell surface receptors might be the candidates regulating the differentiation or gene expressions of mesenchymal cells as amelogenin receptors.

## Conclusion

Our study reveals the novel role of enamel matrix proteins in bone homeostasis and prompts further studies to analyze specific functions of amelogenin spliced variants in bone biology.

## Supporting information

**S1 Fig. Phenotypic comparisons of WT and TgLRAP mice.** A, Alizarin red /alcian blue double staining of 5-day-old calvaria. No change in bone development was apparent. B, Evaluation for bone formation. The calcein intake was compared between WT and TgLRAP. A white circle with 4 mm diameter was used for region of interest (ROI). Green color shows areas of high osteoblast activity and dark color shows those of low osteoblast activity. No significant difference was observed. AU; arbitrary unit. Each column in graphs represents the mean ± SD. n = 5–6 for each group. N.S.; not significant. C, TRAP staining of tibial metaphysis. The number of osteoclasts in secondary spongiosa was counted and compared between WT and TgLRAP samples. The osteoclast number was not significantly different between the groups. B; bone, BM; bone marrow, Arrows; osteoclasts. Bar = 100 μm. Each column in graphs represents the mean ± SD. n = 5–6 for each group. N.S.; not significant.
(TIF)

**S2 Fig. Quantitative analysis of bone formation and resorption in sham /OVX WT or TgLRAP mice.** A, Representative examples for quantitation of calcein incorporation into the calvaria. B, Representative examples of TRAP staining on Sham /OVX WT or TgLRAP mice femur. TRAP positive cells are indicated by arrows. B, bone; BM, bone marrow. Bar = 50 μm.
(TIF)

**S3 Fig. Serum cytokine levels of bone resorption markers.** A, IL-1β. B, RANKL. C, OPG. Each column in graphs represents the mean ± SD. n = 4–6 for each group. N.S.; not significant.
(TIF)

**S4 Fig. Raw images for the Fig 1B and 1D.** The first image in S4 depicts a raw image for Fig 1B and next three images depict raw images for Fig 1D 3 subpanels.
(TIF)

**S1 Table. Sequences for target gene primers.**
(PDF)

## Acknowledgments

We thank the late Dr. Yoshihiko Yamada and Drs. Satoru Takahashi, Wataru Sonoyama, Junko Hatakeyama, Taduru Sreenath, and Kenn Holmbeck for their helpful discussions.

## Author Contributions

**Conceptualization:** Naoto Haruyama, Takayoshi Yamaza, Carolyn W. Gibson, Ashok B. Kulkarni.

**Data curation:** Naoto Haruyama, Shigeki Suzuki, Bradford Hall.

**Formal analysis:** Naoto Haruyama, Takayoshi Yamaza.

**Investigation:** Naoto Haruyama, Takayoshi Yamaza, Shigeki Suzuki, Bradford Hall, Andrew Cho, Carolyn W. Gibson, Ashok B. Kulkarni.

**Methodology:** Naoto Haruyama, Takayoshi Yamaza, Shigeki Suzuki, Andrew Cho.

**Project administration:** Ashok B. Kulkarni.

**Resources:** Ashok B. Kulkarni.

**Supervision:** Ashok B. Kulkarni.

**Validation:** Naoto Haruyama.

**Visualization:** Naoto Haruyama.

**Writing – original draft:** Naoto Haruyama.

**Writing – review & editing:** Takayoshi Yamaza, Shigeki Suzuki, Bradford Hall, Andrew Cho, Carolyn W. Gibson, Ashok B. Kulkarni.

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
