## [Decision Letter · Decision Letter 0]

13 May 2021

PONE-D-21-06528

Leucine rich amelogenin peptide prevents ovariectomy-induced bone loss in mice

PLOS ONE

Dear Dr. Kulkarni,

Thank you for submitting your manuscript to PLOS ONE. After careful consideration, we feel that it has merit but does not fully meet PLOS ONE’s publication criteria as it currently stands. Therefore, we invite you to submit a revised version of the manuscript that addresses the points raised during the review process.

We look forward to receiving your revised manuscript.

Kind regards,

Chi Zhang

Academic Editor

PLOS ONE

Journal Requirements:

" The funders had no role in study design, data collection and analysis, decision to publish, or preparation of the manuscript."

Reviewers' comments:

Reviewer's Responses to Questions

**Comments to the Author**

1. Is the manuscript technically sound, and do the data support the conclusions?

Reviewer #1: Yes

Reviewer #2: Partly

Reviewer #3: Yes

2. Has the statistical analysis been performed appropriately and rigorously? 

Reviewer #1: Yes

Reviewer #2: No

Reviewer #3: Yes

3. Have the authors made all data underlying the findings in their manuscript fully available?

Reviewer #1: Yes

Reviewer #2: No

Reviewer #3: Yes

4. Is the manuscript presented in an intelligible fashion and written in standard English?

Reviewer #1: Yes

Reviewer #2: No

Reviewer #3: Yes

5. Review Comments to the Author

Reviewer #1: Comments

1. In osteoclastogenesis assays, one experiment done for 9 days and the other for 7 days, please provide reasoning for the same. (Line 149-167)

2. Line 208-209, what temperature was incubation performed at?

3. 231-241, please elaborate on the methods used to determine bv/tv,bmd.

4. For Fig 5 (A), it is difficult to differentiate between both the OVX mice. Same with figure B. It might be useful to add a histology section of the same area for comparison. That way the reader can make more conclusive judgments about trabecular bone.

5. Line 239, was BMD of cortical+trabecular bone calculated or only trabecular bone?

6. Please explain why region just under growth plate was chosen and why wasn’t it a larger area.

Reviewer #2: This manuscript describes progress into the inquiry for the role of enamel matrix proteins to moonlight as regulators of both bone formation and resorption. The manuscript describes the creation of a transgenic animal over-expressing an amelogenin isoform, the leucine rich amelogenin peptide, known as LRAP, whose expression was placed under the control of a 2.4 Kbp alpha1(I) collagen promoter. Data are derived from cell and organs of transgenic experimental or littermate control from F2, F3 and F4 mouse generations. Three founder lines are utilized each with varying transgene copy number of 3, 2 and 1 integrations, as well as varying gene expression levels from long bones or skull bones. Line T3 harbored 2 copy number of the transgene compared to the line F14 which carried 3, yet the relative mRNA expression level in T3 bones was 3-6 times higher than controls samples. Remarkably, the expression level of the LRAP transgene in the calvaria or long bone of line T3 was 10000 or 6000 times greater than that in the littermate controls.

Animal sources for all analysis other than ovariectomy are uniformly reported with ambiguity to sex, despite the claim on page 5 line 129 that "Hemizygote transgenic mice (TgLRAP) overexpressing LRAP" suggesting integration was on one or both of the sex chromosomes. How was gender considered as a variable in these experiments?

The manuscript would benefit from editing for conventional American English with an eye to remove laboratory slang, such as "digested out from" (page 5, perhaps changing to "released by restriction endonuclease digestion")" and "skin under" (page 6,perhaps changing to "visualized by") as well as other sites in the manuscript, the use of prepositions. verb tense and voice. Similarly, it is suggested that "greater" be used in describing a number rather than "higher". Please correct the "Acknowledgements" and dedicate the manuscript, if desired, to Dr. Yamada. Indeed Dr. Yamada is missed in the community. If Dr. Yamada made contributions sufficient to be included as an author, this would be appropriate.

The figure legends should decode the data shown without the need for the reader to reference "Materials and Supplies" or "Results". Figure 1 is used as an example, but it would be useful to ensure that the legends address suitable issues, are consistent in sample source and do not require revisiting the manuscript. All of the figure legends confused or slowed the review, selected examples are:

Figure 1.

Panel D line 435. Are these age-, genetic matched littermates? Are all the tissues from Wt or transgenic animals?

Panel C, line 434 "Only bone tissue and teeth....." Indeed, these are the only tissues shown but does not demonstrate the claim of restricted tissue expression.

Line 537 "......expression pattern described in previous reports". Generally, conclusions are not provided in figure legends, but if there is a need, then there is also a need for a reference.

Panel E and F. Transgene expression is dramatically different in the calvaria or perhaps the columns are mislabelled or perhaps the founder line is mislabelled? In either case, some discussion to the biological significance of this is worthwhile.

Figure 2.

The antibody detection levels are sparse, but the transcript number is numerous. Wild type teeth demonstrate a strong amplicon for LRAP, a signal that rivals the 6X larger amelogenin amplicon. These findings are especially relevant to the authors interpretation of the mechanism of action for either stimulating osteogenesis or inhibiting osteoclastogenesis.

Is there a known physiologic level for LRAP to induce bone formation or to inhibit osteoclastogenesis? These issues should be a focus in the Discussion.

The cell origin as an osteocyte is stated to be based on morphology, but no bright field images are provided.

The source of the anti-amelogenin antibody is not given. Other work has suggested that most polyclonal antibodies to the full length amelogenin (M180) are not effective at detecting the smaller LRAP molecule in either tissue sections or Western assays.

Please consider providing the rationale for using a collagen promoter rather than a promoter restricted only to osteoblasts or osteocytes or even to the bone marrow.

The technical aspect of the manuscript moves freely between "hind limb bones", the femur and the parietal calvarial bone plate. Please provide a rationale for why each of these organs can be considered equivalent. Is this in consideration that the parietal bone is paraxial mesoderm rather than neural crest?

Is there data describing the chosen markers for bone turnover as described on page 9, lines 212-229?

Page 13, lines 304-305. There is no data to support the claim that LRAP is secreted.

Which is the more dominant effect of exogenous LRAP--anabolic bone formation or catabolic osteoclastogenesis? Can this be deciphered with this data?

Page 14 lines 332 to 334 are perplexing. It states, "To understand the reason for suppressed osteoclasts formation, cytokines closely related to osteoclastogenesis were measured by ELISA. The systemic changes of serum RANKL, OPG, TNF-alpha, IL-1beta, or IL-6 were not observed in this series of experiments (data not shown)." So--should these line of investigation even been mentioned? It seems that these markers should have been measured and the authors are aware of this need. It is not clear what is meant from this wording.

The work of other scientists to elucidate the role of enamel matrix proteins in bone physiology is shallow and does not appropriately reflect the impact that primate and human studies on bone and cementum regeneration by a commercial product Emdogain (enamel matrix proteins including LRAP) had on stimulating research in this area.

The work of Lyngstadaas and co-workers to decipher amelogenin signalling is absent.

Interest in enamel matrix proteins on bone came not due solely to the creation of a null animal as presented.

However, the manuscript does capture the contributions of Veis and colleagues who suggested that amelogenin proteins could play additional roles by inducing dentinogenesis.

Zhou and colleagues have also provided published data supporting a canonical Wnt signalling role for LRAP during induction of murine or human marrow stromal cells to osteogenic phenotypes. The importance of the osteogenic pathway plays to scanty of a role in interpreting the findings in this manuscript.

Reviewer #3: The present manuscript by Drs. Kulkarni and colleagues characterizes the effects of ovariectormy-induced bone loss in mice. This is a well written and well documented manuscript that highlights the effects of an amelogenin splice product on bone homeostasis. Specifically, the authors demonstrate that LRAP overexpressing mice prevent osteoclast activity in ovariectomized mice. Overall, the study is well executed and spans from in vitro data to gene expression data in calvarial cells.

Based on this background it is surprising that the authors have not quoted and discussed a previous study on the "Expression and Function of Enamel-related Gene Products in Calvarial Development" by Drs. Atsawasuwan and colleagues in the Journal of Dental Research (2013), especially since this paper makes a lot of sense in the context of previous manuscripts related to the effects of proteins previously only associated with the enamel layer in bone. This publication should be included upon revision.

Otherwise this is a very fine expertly conducted study.

6. PLOS authors have the option to publish the peer review history of their article (what does this mean?). If published, this will include your full peer review and any attached files.

Reviewer #1: No

Reviewer #2: No

Reviewer #3: No

---

## [Author Response · Author response to Decision Letter 0]

28 Sep 2021

Response > We have checked to ensure that our manuscript meets PLOS ONE's style requirements.

2. Thank you for stating the following financial disclosure: "The funders had no role in study design, data collection and analysis, decision to publish, or preparation of the manuscript. "At this time, please address the following queries:

a. Please clarify the sources of funding (financial or material support) for your study. List the grants or organizations that supported your study, including funding received from your institution.

d. If you did not receive any funding for this study, please state: “The authors received no specific funding for this work.”

Response > We have added amended statements in our new cover letter.

Response > The file for raw data named ‘S4_raw_images.pdf’ was uploaded as a Supporting Information File, which is also noted in our new cover letter. 

Response > The new figure file was uploaded as a Supporting figure: S3 Fig. The relevant lines at the end of Results and the second paragraph of Discussion sections have been revised. 

Reviewer's Responses to Questions

Comments to the Author

1. Is the manuscript technically sound, and do the data support the conclusions? 

Response > We have revised the entire manuscript. We hope that the reviewers will find the manuscript technically sound, and the data supporting our conclusions.

2. Has the statistical analysis been performed appropriately and rigorously? 

Response > We have checked the methods for statistical analysis again. We believe that we have utilized appropriate and rigorous statistical analysis.

3. Have the authors made all data underlying the findings in their manuscript fully available? 

Response > We have uploaded all the raw data and data underlying the findings as Supporting Information files.

4. Is the manuscript presented in an intelligible fashion and written in standard English? 

Response > We have corrected all typographical or grammatical errors for better English and edited the manuscript by eliminating the lab slang.

5. Review Comments to the Author

Reviewer #1: Comments

1. In osteoclastogenesis assays, one experiment done for 9 days and the other for 7 days, please provide reasoning for the same. (Line 149-167)

Response > We are thankful to this reviewer for his/her valuable comments. Generally, direct induction of osteoclastogenesis from the spleen requires additional several days as compared to the osteoclastogenesis in the coculture system since the mesenchymal cells such as calvarial cells or bone marrow stromal cells as a source of M-CSF/RANKL do not exist in the direct induction of osteoclastogenesis system. Please see Iris Boraschi-Diaz and Svetlana V. Komarova, Cytotechnology. 2016 Jan; 68(1): 105–114. doi: 10.1007/s10616-014-9759-3.

2. Line 208-209, what temperature was incubation performed at?

Response > We provided the temperature for incubation in the lines.

3. 231-241, please elaborate on the methods used to determine bv/tv, bmd.

Response > We elaborated the methods for measurements for bone parameters as follows. ‘The femora were further used for three-dimensional (3D) micro-computed tomographic (µCT) analyses (eXplore MS, GE Medical Systems). The femora and a calibration phantom (SB3, Gammex RMI) were scanned with 8-µm isotropic voxels (80 kV, 80 µA) on the µCT system and the reconstructed 3D images of distal femora were analyzed using MicroView software (GE Medical Systems). The trabecular (cancellous) bone volume/ tissue volume ratio (BV/TV; %) and the trabecular bone mineral density (BMD; mgHA/cm3) as hydroxyapatite (HA) equivalents in a cylindrical area (ROI; Ø1.0 x 0.5 mm) (Fig. 5B) within the distal metaphyseal area underneath the growth plate-metaphyseal junction were analyzed.’ Accordingly, we modified the units in Fig. 5C from mg/cm3 to mgHA/cm3.

4. For Fig 5 (A), it is difficult to differentiate between the OVX mice. Same with figure B. It might be useful to add a histology section of the same area for comparison. That way the reader can make more conclusive judgments about trabecular bone.

Response > According to the comments, we replaced the left panels in Fig. 5B with the 3D models at the ROI level. Each panel of Fig. 5A was also magnified. Now we believe that the phenotypic difference of the trabecular bone between OVX WT and TgLRAP is clearly demonstrated. 

5. Line 239, was BMD of cortical+trabecular bone calculated or only trabecular bone?

Response > The BMD of trabecular bone in a cylindrical area (ROI; Ø1.0 x 0.5 mm) was measured. That is now clearly indicated in the Method according to the modifications based on your comment #3.

6. Please explain why region just under growth plate was chosen and why wasn’t it a larger area. 

Response > We tried to investigate the bone remodeling, rather than bone modeling characteristics. To analyze the bone remodeling, trabecular bone is most frequently assessed at the distal femoral metaphysis. We added a new reference #42 for the selection of ROI at the end of ‘Radiographic and micro computed tomography (µCT) analysis’ section. 

Reviewer #2: 

Animal sources for all analysis other than ovariectomy are uniformly reported with ambiguity to sex, despite the claim on page 5 line 129 that "Hemizygote transgenic mice (TgLRAP) overexpressing LRAP" suggesting integration was on one or both sex chromosomes. How was gender considered as a variable in these experiments? 

Response > We are thankful to this reviewer for his/her helpful comments. Although we did not identify the locus of transgene’s integration, we assume that the transgene was integrated into autosome since the transgenic founder mice except for mosaic line (T7) inherited the transgene at 50% probability. Because the transgene integrates randomly into mouse genomes, there would be a possibility to disrupt a gene expression especially if the mice were maintained as a homozygote. Therefore, we maintained the transgenic mice as ‘heterozygote’. However, if one transgene allele is missing, it should be indicated as ‘hemizygote’ for the transgenic mice. Anyway, female mice were used for the analysis except for the mice at 3- and 5-day-old, when the sex was difficult to be determined. This sex information is now indicated in the 2nd paragraph of the new ‘Generation of LRAP transgenic mice and genotyping’ section.

The manuscript would benefit from editing for conventional American English with an eye to remove laboratory slang, such as "digested out from" (page 5, perhaps changing to "released by restriction endonuclease digestion")" and "skin under" (page 6, perhaps changing to "visualized by") as well as other sites in the manuscript, the use of prepositions. verb tense and voice. Similarly, it is suggested that "greater" be used in describing a number rather than "higher". 

Response > We have improved the manuscript for better English. We have eliminated the lab slang.

Please correct the "Acknowledgements" and dedicate the manuscript, if desired, to Dr. Yamada. Indeed Dr. Yamada is missed in the community. If Dr. Yamada made contributions sufficient to be included as an author, this would be appropriate.

Response > Thank you for your suggestion, but we feel such dedication will be appropriate in a forth coming paper from his lab or his former fellows. We would like to maintain the Acknowledgment for the late Dr. Yoshihiko Yamada for his gift of reagents.

The figure legends should decode the data shown without the need for the reader to reference "Materials and Supplies" or "Results". Figure 1 is used as an example, but it would be useful to ensure that the legends address suitable issues, are consistent in sample source and do not require revisiting the manuscript. All the figure legends confused or slowed the review, 

Response > We have elaborated the legends for all figures. 

selected examples are:

Figure 1.

Panel D line 435. Are these age-, genetic matched littermates? Are all the tissues from Wt or transgenic animals?

Response > We changed the figure legend for panel D as indicated below. “Screening for transgene expression in major organs. RT-PCR was performed to detect transgene expression in various tissues from transgenic mice (TgLRAP) at 5-day-old. The transgene expression was predominantly detected in teeth, bone, and tail tissue. WT teeth; positive and negative control sample from a littermate for the expression of amelogenin and IRES-EGFP (transgene), respectively. GAPDH; glyceraldehyde 3-phosphate dehydrogenase as an internal control.” We have also changed the labels in Fig. 1D to clearly indicate the sample source.

Panel C, line 434 "Only bone tissue and teeth....." Indeed, these are the only tissues shown but does not demonstrate the claim of restricted tissue expression.

Response > We have removed ‘Only’ in this sentence. We have also replaced the ‘Only’ with ‘predominantly’ in the 1st paragraph of ‘Results’ section.

Line 537 "......expression pattern described in previous reports". Generally, conclusions are not provided in figure legends, but if there is a need, then there is also a need for a reference.

Response > We deleted the following lines from the legends, ‘which is consistent with the expression pattern described in previous reports’

Panel E and F. Transgene expression is dramatically different in the calvaria or perhaps the columns are mislabeled or perhaps the founder line is mislabeled? In either case, some discussion to the biological significance of this is worthwhile.

Response > We have added the following sentence for this discrepancy between the copy numbers of transgene and mRNA expression level in the figure legend for panel E. ‘Although line T3 harbored a lower copy number compared to line F14, the relative mRNA expression level in calvaria was ~6 times higher than that in line F14, possibly due to the difference of integration locus of the transgene.’ 

Figure 2.

The antibody detection levels are sparse, but the transcript number is numerous. Wild type teeth demonstrate a strong amplicon for LRAP, a signal that rivals the 6X larger amelogenin amplicon. These findings are especially relevant to the authors interpretation of the mechanism of action for either stimulating osteogenesis or inhibiting osteoclastogenesis.

Is there a known physiologic level for LRAP to induce bone formation or to inhibit osteoclastogenesis? These issues should be a focus in the Discussion.

Response > As the reviewer rightly pointed out, the detection level of LRAP on the TgLRAP tissue sections appeared to be lower than the expected level based on the numerous transcript number of TgLRAP (Fig. 2). The reason for the low detection level of LRAP by antibody would be that the secreted protein is generally detected at a relatively lower level in immunohistochemistry as compared to other proteins confined in the cytoplasm. We do not know the physiological or minimal level for LRAP to induce bone formation or to inhibit osteoclastogenesis. However, the endogenous expression level of amelogenin is enough to change the osteoblastic activity of ES cells (ref#35) and calvarial osteoblasts (ref#24) or the osteoclastogenesis by PDL cells (ref#28). Therefore, we believe that the amount of LRAP should be above the biologically effective dose since the expression level of amelogenin protein in the TgLRAP bone tissues was higher than the WT control (Fig. 2). The discussion above is now added to the new 3rd paragraph of Discussion. 

The cell origin as an osteocyte is stated to be based on morphology, but no bright field images are provided.

Response > Technically, it is difficult to show the bright field images since the IHC was performed using fluorescence antibodies. We changed the statement ‘osteocyte’ to the ‘osteocyte-like cells located in the bone matrix’. We also modified the corresponding lines in Discussion according to this change.

The source of the anti-amelogenin antibody is not given. Other work has suggested that most polyclonal antibodies to the full length amelogenin (M180) are not effective at detecting the smaller LRAP molecule in either tissue sections or Western assays.

Response > Amelogenin antibody corresponding to the amelogenin carboxy-terminus was used for our experiments. We provided reference #41 for the source of amelogenin antibody. We do not know the appropriate mouse tissues in which endogenous LRAP is strongly but exclusively expressed, other than the tissues from our TgLRAP mice. However, we have already shown that the antibody recognized both the M180 and LRAP at least in Western assays (ref #41 and Fig. 1 in Mitani et al. Oral Dis. 2013 Mar;19(2):169-79. PMID 22863294). 

Please consider providing the rationale for using a collagen promoter rather than a promoter restricted only to osteoblasts or osteocytes or even to the bone marrow.

Response > As previously reported (ref #37, 46), transgenic mice harboring 0.9 kb of the type I collagen promoter expressed the transgene at relatively low levels almost exclusively in the skin. On the other hand, mice with a sequence containing 2.3 kb of this proximal promoter expressed the transgene at high levels in osteoblasts (or odontoblasts or some osteocytes) but not in other type I collagen–producing cells. Transgenic mice harboring 3.2 kb of the proximal promoter showed an additional high-level expression of the transgene in tendon and fascia fibroblasts. The rationale for using the 2.3kb collagen promoter is now clearly provided at the end of 2nd paragraph of the ‘Discussion’.

The technical aspect of the manuscript moves freely between "hind limb bones", the femur and the parietal calvarial bone plate. Please provide a rationale for why each of these organs can be considered equivalent. Is this in consideration that the parietal bone is paraxial mesoderm rather than neural crest?

Response > We understand that the parietal bone and femur (hind limb bones) are derived from the same paraxial mesoderm. The reason why we analyzed both the calvaria and hind limb long bones was that we tried to examine the phenotypic and expressional differences between bones that ossify with membranous ossification (parietal bone) or endochondral ossification (hind limbs) since some mouse models show different phenotypic changes in each bone according to the expression levels. For example, heterozygous Cbfa1+/- embryos exhibited phenotypic changes predominantly in clavicles and calvarial bones (ref #44). Here, we identified that the amelogenin expression level in the calvarial bone and the hind limb long bones was different (Fig. 1F). However, there were no obvious phenotypic changes in the calvarial and hind limb bones at least in non-OVX mice. To be clear the reviewer’s point, we added the sentence above in ‘TgLRAP mice resisted bone loss induced by OVX’ section of the Results.

Is there data describing the chosen markers for bone turnover as described on page 9, lines 212-229?

Response > Upon revision, we omitted several markers for bone turnover since we found that we had only the preliminary data for several markers. The selected data have been provided as a Supporting figure; S3 Fig, in the revised manuscript.

Page 13, lines 304-305. There is no data to support the claim that LRAP is secreted.

Response > We replaced the sentence ‘which secreted the LRAP’ with ‘obtained from TgLRAP mice’.

Which is the more dominant effect of exogenous LRAP--anabolic bone formation or catabolic osteoclastogenesis? Can this be deciphered with this data?

Response > Unfortunately, we do not have enough data to decipher the dominant effect of exogenous LRAP on anabolic or catabolic effects. However, we believe that the LRAP affects both the anabolic and catabolic aspects of bone metabolism to a certain extent as shown in this paper.

Page 14 lines 332 to 334 are perplexing. It states, "To understand the reason for suppressed osteoclasts formation, cytokines closely related to osteoclastogenesis were measured by ELISA. The systemic changes of serum RANKL, OPG, TNF-alpha, IL-1beta, or IL-6 were not observed in this series of experiments (data not shown)." So--should these line of investigation even been mentioned? It seems that these markers should have been measured and the authors are aware of this need. It is not clear what is meant from this wording.

Response > We have reworded the sentence as follows. "To understand the reason for suppressed osteoclasts formation, serum cytokine levels including IL-1beta, RANKL, and OPG, those are closely related to osteoclastogenesis, were measured by ELISA. However, none of those cytokine levels showed significant differences between the experimental groups (S3 Fig. A-C)." Upon revision, we omitted several markers for bone turnover since we found that we had only the preliminary data for several markers.

The work of other scientists to elucidate the role of enamel matrix proteins in bone physiology is shallow and does not appropriately reflect the impact that primate and human studies on bone and cementum regeneration by a commercial product Emdogain (enamel matrix proteins including LRAP) had on stimulating research in this area. The work of Lyngstadaas and co-workers to decipher amelogenin signaling is absent. Interest in enamel matrix proteins on bone came not due solely to the creation of a null animal as presented.

Response > We have added the previous work by Lyngstadaas et al. in the Introduction (ref #33).

However, the manuscript does capture the contributions of Veis and colleagues who suggested that amelogenin proteins could play additional roles by inducing dentinogenesis. Zhou and colleagues have also provided published data supporting a canonical Wnt signaling role for LRAP during induction of murine or human marrow stromal cells to osteogenic phenotypes. The importance of the osteogenic pathway plays to scanty of a role in interpreting the findings in this manuscript. 

Response > We introduced the previous work by Warotayanont et al. published by Zhou’s group (ref #35) in the Introduction. We also modified the 3rd paragraph of Introduction and Discussion so that the importance of bone formation pathway activated by LRAP can be introduced.

Reviewer #3: The present manuscript by Drs. Kulkarni and colleagues characterizes the effects of ovariectormy-induced bone loss in mice. This is a well written and well documented manuscript that highlights the effects of an amelogenin splice product on bone homeostasis. Specifically, the authors demonstrate that LRAP overexpressing mice prevent osteoclast activity in ovariectomized mice. Overall, the study is well executed and spans from in vitro data to gene expression data in calvarial cells.

Based on this background it is surprising that the authors have not quoted and discussed a previous study on the "Expression and Function of Enamel-related Gene Products in Calvarial Development" by Drs. Atsawasuwan and colleagues in the Journal of Dental Research (2013), especially since this paper makes a lot of sense in the context of previous manuscripts related to the effects of proteins previously only associated with the enamel layer in bone. This publication should be included upon revision.

Response > We thank this reviewer for pointing out this important omission. We have now included the suggested paper (ref#24) by modifying the 2nd and 3rd paragraphs of the Introduction. We have also discussed the paper in the 2nd and 3rd paragraphs of Discussion in the revised manuscript.

Otherwise this is a very fine expertly conducted study. 

Response > Thanks for your kind comment.

---

## [Decision Letter · Decision Letter 1]

2 Nov 2021

Leucine rich amelogenin peptide prevents ovariectomy-induced bone loss in mice

PONE-D-21-06528R1

Dear Dr. Kulkarni,

We’re pleased to inform you that your manuscript has been judged scientifically suitable for publication and will be formally accepted for publication once it meets all outstanding technical requirements.

Kind regards,

Chi Zhang

Academic Editor

PLOS ONE

Additional Editor Comments (optional):

Reviewers' comments:

Reviewer's Responses to Questions

**Comments to the Author**

1. If the authors have adequately addressed your comments raised in a previous round of review and you feel that this manuscript is now acceptable for publication, you may indicate that here to bypass the “Comments to the Author” section, enter your conflict of interest statement in the “Confidential to Editor” section, and submit your "Accept" recommendation.

Reviewer #3: All comments have been addressed

Reviewer #4: All comments have been addressed

2. Is the manuscript technically sound, and do the data support the conclusions?

Reviewer #3: (No Response)

Reviewer #4: Yes

3. Has the statistical analysis been performed appropriately and rigorously? 

Reviewer #3: (No Response)

Reviewer #4: Yes

4. Have the authors made all data underlying the findings in their manuscript fully available?

Reviewer #3: (No Response)

Reviewer #4: Yes

5. Is the manuscript presented in an intelligible fashion and written in standard English?

Reviewer #3: (No Response)

Reviewer #4: Yes

6. Review Comments to the Author

Reviewer #3: (No Response)

Reviewer #4: (No Response)

7. PLOS authors have the option to publish the peer review history of their article (what does this mean?). If published, this will include your full peer review and any attached files.

Reviewer #3: No

Reviewer #4: No

---

## [Editor Report · Acceptance letter]

5 Nov 2021

PONE-D-21-06528R1 

Leucine rich amelogenin peptide prevents ovariectomy-induced bone loss in mice 

Dear Dr. Kulkarni:

I'm pleased to inform you that your manuscript has been deemed suitable for publication in PLOS ONE. Congratulations! Your manuscript is now with our production department. 

Kind regards, 

on behalf of

Dr. Chi Zhang 

Academic Editor

PLOS ONE